# Development and validation of a portable, point-of-care canine distemper virus qPCR test

Ania Tomaszewicz Brown[1], Denise McAloose[1], Paul P. Calle[1], Angelika Auer[2], Annika Posautz[3], Sally Slavinski[4], Robin Brennan[5], Chris Walzer[3,6], Tracie A. Seimon[1]*

1 Wildlife Conservation Society, Zoological Health Program, Bronx, New York, United States of America, 2 Institute of Virology, University of Veterinary Medicine, Vienna, Austria, 3 Research Institute of Wildlife Ecology (FIWI), University of Veterinary Medicine, Vienna, Austria, 4 New York City Department of Health and Mental Hygiene, Queens, New York, United States of America, 5 Animal Care Centers of New York, New York, New York, United States of America, 6 Wildlife Conservation Society, Bronx, New York, United States of America

* tseimon@wcs.org

**Data Availability Statement:** All relevant data are within the manuscript and its Supporting Information files.

## Abstract

Canine distemper virus (CDV) is a multi-host pathogen that can cause significant mortality in domestic, wild terrestrial and marine mammals. It is a major conservation threat in some endangered species. Infection can result in severe respiratory disease and fatal encephalitis. Diagnosis and disease monitoring in wildlife, and differentiation of CDV from rabies (a life-threatening zoonotic disease that can produce similar neurologic signs), would benefit from the availability of a portable, point-of-care (POC) diagnostic test. We therefore developed a quantitative RT-PCR assay for CDV using shelf-stable, lyophilized reagents and target-specific primers and probes for use with the handheld Biomeme two3™ qPCR thermocycler. Biomeme's extraction methodology, lyophilized reagents, and thermocycler were compared to our standard laboratory-based methods to assess sensitivity, efficiency and overall test performance. Results using a positive control plasmid for CDV showed comparable sensitivity (detection of 50 copies) and PCR efficiency between the two platforms, and CDV detection was similar between platforms when tested using a modified live CDV vaccine. Significantly higher Ct values (average Ct = 5.1 cycles) were observed using the Biomeme platform on known CDV positive animal samples. CDV detection using the Biomeme platform was similar in 25 of 26 samples from suspect CDV cases when compared to standard virology laboratory testing. One false positive was observed that was negative upon retest. The Biomeme methodology can be adapted for detection of specific targets, and this portable technology saves time by eliminating the need for local or international sample transport for laboratory-based diagnostics. However, results of our testing suggest that decreased diagnostic sensitivity (higher Ct values) relative to laboratory-based methods was observed using animal samples, so careful validation and optimization are essential. Portable qPCR platforms can empower biologists and wildlife health professionals in remote and low-resource settings, which will greatly improve our understanding of CDV disease ecology and associated conservation threats in wildlife.

**Funding:** The author(s) received no specific funding for this work.

**Competing interests:** The authors have declared that no competing interests exist.

## Introduction

Canine distemper virus (CDV), a *Morbillivirus* in the *Paramyxoviridae* family, is a multi-host pathogen that is found globally and has a wide host-range. All members of the order Carnivora are thought to be susceptible, and CDV exposure has also been described in several species of rodents, primates, suids, cervids, and elephants [1]. Significant mortality events have been described in terrestrial and marine mammals [2], and disease and outbreaks in endangered African wild dogs (*Lycaon pictus*) [3], Ethiopian wolves (*Canis simensis*) [4], Amur tigers (*Panthera tigris altaica*) [5–7], Amur leopards (*Panthera pardus orientalis*) [8], Iberian lynx (*Lynx pardinus*) [9], and black-footed ferrets (*Mustela nigripes*) [10] are considered potentially significant conservation threats to these species. Domestic dogs and mesocarnivores are known reservoir hosts [2,6,7]. The virus is epitheliotropic, lymphotropic and neurotropic, and clinical signs are often seen in the respiratory, lymphoid, gastrointestinal and nervous systems [11]. Common outcomes of infection include lymphoid depletion, hyperkeratosis, interstitial pneumonia (often complicated by opportunistic bacterial infections), and fatal encephalopathy.

Portable, point-of-care (POC) technologies allow scientists, biologists, medical professionals and the public to take advanced diagnostics to the field [12]. Though not a new concept, few POC tools exist for domestic companion animal or livestock diagnostics. Fewer have been developed and validated for use in wildlife and field settings [13]. However, recent outbreaks of Ebola in great apes and humans, SARS in civets, *peste-des-petits* ruminants in saiga, African swine fever, MERS in camels and humans, and many other diseases of animal and public health importance have highlighted a need for development of new or enhancement of existing portable POC tools [12,14–20].

POC diagnostics can help address many challenges in understanding CDV presence, transmission patterns, identification of disease outbreaks, and conservation threats in wildlife. Some of these challenges include access to animals and opportunistic testing across small and large geographic ranges, low density or elusive behavior of some target species, and absence of or limited monitoring efforts. Others challenges researchers face include limited expertise necessary for appropriate animal sample collection, handling, and storage (including maintaining a cold chain), absence of available laboratory testing and differentiating from disease such as rabies which can present with similar clinical manifestations, and/or logistical challenges related to permit processes needed for regional or international sample shipping for laboratory testing. These challenges are compounded in remote and/or low resource settings. POC diagnostics are increasingly providing opportunities for rapid testing by researchers while they are already collecting data in the field, or when handling sick or dead wildlife, and with the development and validation of more user-friendly kits, researchers have opportunities to overcome many of these obstacles and logistical challenges that would normally impede testing.

Current methods for CDV detection include histology, electron microscopy, immunohistochemistry (IHC), enzyme-linked immunosorbent assay (ELISA), virus isolation, and conventional and quantitative reverse-transcription PCR (RT-qPCR). All of these methods are generally performed in standard diagnostic laboratories using stationary bench-top analyzers run by trained professionals. While working in the field, limited access to this infrastructure or access to portable diagnostics can negatively impact the accessibility of timely results, case identification, and development of strategies to mitigate disease transmission and spread.

The goal of this study was to develop and validate a rapid, portable, field-friendly, CDV-specific, POC RT-qPCR test for wildlife or domestic animal diagnostics. To do so, we compared the sensitivity and efficiency of each component of the Biomeme POC platform (biomeme. com) to our standard, laboratory-based RT-qPCR platform. We subsequently compared the

performance of these two platforms using samples from free ranging wildlife in the United States and Austria that were collected during recent natural CDV outbreaks in wild carnivores.

The Biomeme platform includes the M1 Sample Prep Kit™ for RNA extraction, LyoRNA™ RT-PCR mastermix with custom primers and a TaqMan probe, and their two3™ thermocycler [13]. The Biomeme M1 Sample Prep Kit™ for RNA extraction is pre-packaged and uses a syringe-mounted extraction column (Fig 1, top). Through the use of four color-coded reagents, the extraction process easily overcomes training and language barriers. The Biomeme LyoRNA™ mastermix is lyophilized and shelf-stable. It can be combined with lyophilized primers and probes into a bead that is pre-packaged into Biomeme qPCR reaction tube 'Go-Strips™'. Reagents, primers and probes can be manufactured to meet specific needs. Reconstitution of the beads with DNA or RNA template prepares the sample for PCR. The Biomeme two3™ qPCR machine (Biomeme Inc. Philadelphia, PA, USA) is a small, light-weight, portable thermocycler that can be hand-held and displays output through a smart phone or laptop-based application [13]. All of the above steps are performed without the need for heat blocks, centrifuges, or cold storage, and the thermocycler can be run on battery or solar power.

## Materials and methods

### Ethics statement

All samples were collected opportunistically from animals that were found dead or humanely euthanized because of severe illness during naturally occurring wildlife disease outbreaks. Following AVMA Euthanasia Guidelines Raccoons were sedated with intramuscular ketamine prior to the IV or IP injection of sodium pentobarbital. Standard techniques for animal handling, euthanasia, and sampling were performed by licensed veterinarians and technicians in accordance with local, regional, national and international guidelines/best practices and laws. IACUC review and approval for the wildlife handling and sample collection performed in this project are not required in New York State, USA, or in the European Union. No live protected species were handled or sampled for this project.

### RNA extraction

**RNA extraction using the Biomeme RNA field prep kit.**   RNA extraction from reconstituted Duramune™ MAX 5 Canine Distemper-Adenovirus Type 2-Parainfluenza-Parvovirus (DA2PP) modified live virus vaccine (Elanco US Inc., Fort Dodge, IA), hair, swabs of tissue (fresh frozen or RNA*later*™ preserved), and fresh frozen tissue from suspected CDV positive animals was carried out at both at the Zoological Health Program's Molecular Laboratory at the Bronx Zoo, New York, NY, USA and University of Veterinary Medicine, Vienna, Austria. These samples were extracted using the Biomeme RNA M1 Sample Prep Kit™. Samples (100 μL of a 1:10 vaccine dilution series, swab, hair or tissue) were placed in 1 mL of BLB (Biomeme lysis buffer) with 5 μL of carrier RNA (Qiagen Inc., CA, USA). Some samples were extracted with 1mL BLB plus 500 μL of molecular grade ethanol to try and enhance RNA viral recovery, however we did not find the addition of ethanol had any effect or improved viral recovery so was eliminated from this protocol early in the study. The samples were vigorously shaken by hand for one minute and incubated at room temperature for 10 minutes. The entirety of the lysate was then drawn up into the Biomeme syringe through the extraction column and gently pumped up and down through the column 10 times, expelling, then discarding the liquid on the final pump. Next, 400 μL of BWB (Biomeme wash buffer) was drawn up into the column and expelled (1 pump). Then 800 uL of BPW (Biomeme protein wash) was drawn up in the column and expelled (1 pump). To eliminate excess fluid in the column, air was drawn up into the syringe and was then completely expelled by pumping residual fluid into a small waste

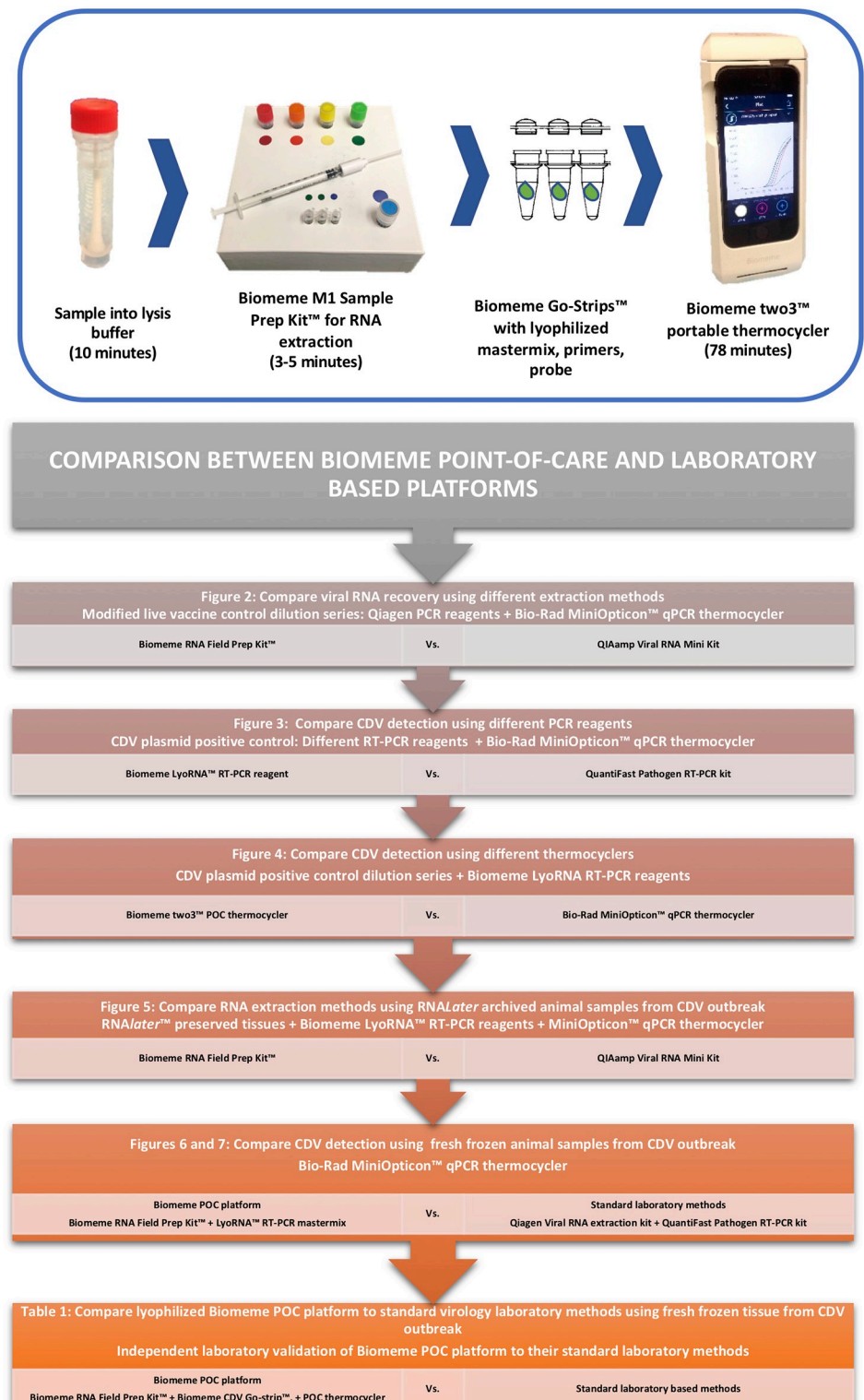

**Fig 1. Biomeme POC platform components, and flow-chart of experiments conducted.** Top. Photographic depiction of the step-wise protocol of Biomeme platform including Lysis step, RNA extraction step using the M1 Sample Prep Kit™, PCR reaction using the Biomeme Go-Strips, and qPCR step using the Biomeme two3™ thermocycler. Bottom. Flow chart of experiments conducted to validate the Biomeme platform and compare it to standard benchtop platform, with reference to corresponding figures where each step was conducted.

container repeatedly approximately 10 times to dry the column. RNA was eluted from the column by drawing up 250 μL of BEB (Biomeme elution buffer) then pumping it up and down through the column three times. The eluate was expelled into an RNase/DNase-free tube.

**RNA extraction using the Qiagen RNA kits.** RNA extractions from reconstituted DA2PP vaccine, hair samples, and swabs of fresh frozen or RNA*later*™ preserved tissue from CDV positive animals was performed using the QIAamp® Viral RNA Mini Kit (Qiagen Inc., CA, USA). Fresh frozen tissue was extracted using the AllPrep DNA/RNA Mini Kit (Qiagen Inc., CA, USA). Qiagen extraction procedures followed the manufacturer's protocol and the final extract was eluted with 60 μL DNAse and RNAse-free water for all samples.

**RNA extractions at the Institute of Virology, Vienna.** For samples tested at the Institute of Virology, University of Veterinary Medicine in Vienna, Austria, standard laboratory methods were used. RNA was extracted from reconstituted fresh frozen preserved tissue. Briefly, an organ suspension of 100 mg of organ tissue into 1 mL of PBS was made. The suspension was then lysed using 3–4 steel beads in a TissueLyser II (Qiagen Inc., Hilden, Germany). The sample was then centrifuged for one minute at 13,000 rpm, and the clarified lysate was extracted using QIAamp® Viral RNA Mini Kit in conjunction with a QIAcube (Qiagen Inc., Hilden, Germany) following the manufacturer's instructions.

## Quantitative real-time PCR

All samples were tested in singlicate except in experiments where we could accommodate duplicate or triplicate replicates on the plate where indicated.

**PCR conditions using Biomeme LyoRNA™ RT-PCR mastermix reagents.** Samples were tested using quantitative, reverse transcriptase PCR (RT-qPCR) to amplify a 114 bp region of the phosphoprotein (P) gene of canine distemper virus as previously described [21]. Each 25 μL PCR reaction contained the following primers and probe: CDVF4, GTCGGTAATCGAG–GATTCGAGAG and CDVR3, GCCGAAAGAATATCCCCAGTTAG, (0.4 μM working concentration), CDV MGB Taqman probe 6FAM-ATCTTCGCCAGAATCCTCAGTGCT-MGBNFQ (0.2 μM working concentration) (Thermo Fisher Scientific, MA, USA). In addition to the primers and probe, each 25 μL reaction contained 2 μL of RNA template, 17.5 μL of DNase/RNase-free water and 5 μL of reconstituted lyophilized Biomeme LyoRNA™ RT-PCR mastermix (5X concentration). Samples using Biomeme LyoRNA™ mastermix were tested using either the Biomeme two3™ or the Bio-Rad MiniOpticon™ thermocycler under the following cycling conditions: 50 ˚C for 2 minutes, 95 ˚C for 1 minute, followed by 45 cycles of 95 ˚C for 15 seconds, 60 ˚C for 30 sections (data collection step).

**PCR using QuantiFast Pathogen RT-PCR mastermix and the Bio-Rad MiniOpticon™ thermocycler.** Samples were tested with the QuantiFast Pathogen RT-PCR Kit and reagents (Qiagen Inc., CA, USA) using RT-qPCR to amplify a 114bp region of the CDV phosphoprotein (P) gene as described above. In addition to the primers and probe concentrations listed above, each 25 μL PCR reaction contained the following: 2 μL of RNA template, 12.5 μL of DNase/RNase-free water, 5 μL of 5x master mix, 0.25 μL of 100X reverse transcriptase, 2.5 μL of 10X IC RNA and 2.5 μL of IC (inhibition control) mix. Samples were tested on the Bio-Rad MiniOpticon™ with the following cycling conditions: 50 ˚C for 20 minutes, 95 ˚C for 5 minutes, followed by 45 cycles of 95 ˚C for 15 seconds, 60 ˚C for 30 seconds (data collection step), and then held at 12 ˚C. A no-template negative control and plasmids containing the primer binding sites for the CDV P gene were included as negative and positive controls, respectively.

**PCR conditions using Biomeme CDV Go-Strips™.** Samples from suspect CDV cases were tested using the Biomeme platform at the Research Institute of Wildlife Ecology, University of Veterinary Medicine, Vienna. 3-well Biomeme CDV Go-Strips™ with lyophilized beads

containing Biomeme LyoRNA™ RT-PCR mastermix were manufactured with the concentrations of target specific primers and probe as above. When using the Biomeme CDV Go-Strips™, the final RNA elution was diluted 1:4 in RNAase-free water, and 20 μL of diluted template was added to the bead for a final total 20 μL PCR reaction volume. Samples using Biomeme CDV Go-Strips™ were tested using the Biomeme two3™ machine under the following cycling conditions: 50 ˚C for 2 minutes, 95 ˚C for 1 minute, followed by 45 cycles of 95 ˚C for 15 seconds, 60 ˚C for 30 sections (data collection step). Samples were tested with both positive and negative controls. A no-template negative control and plasmids containing the primer binding sites for the CDV P gene were included in each RT-PCR experiment.

**PCR using qScript™ XLT One-Step RT-qPCR ToughMix®, ROX™ and Rotor-Gene Q thermocycler.** Samples tested at the diagnostic laboratory of the Institute of Virology, University of Veterinary Medicine, Vienna were tested using 20 μL reactions that contained 2.5 μL of RNA template, 4 μL of DNase/RNase-free water, 10 μL of 2X qScript™ XLT One-Step RTqPCR ToughMix®, ROX™ (Quanta Biosciences™), 2 μL of 50 mM MgSO4, with 0.5 μL of the following primers targeting an 87bp product of the CDV nucleocapsid (N) gene from a 40.0 μM stock concentration; CDV-F: 5−AGCTAGTTTCATCTTAACTATCAAATT −3; CDV-R: 5‵−TTAAC TCTCCAGAAAACTCATGC−3; and 0.5 μL of the following probe from a 20.0 μM concentration; CDV-PROBE: FAM-5−ACCCAAGAGCCGGATACATAGTTTCAATGC−3-TAMRA. PCR cycling conditions were 50˚C for 15 minutes, then 95˚C for 2 minutes, followed by 45 cycles of 95˚C for 15 seconds, 60˚C for 30 seconds (data collection step). RT-qPCR was performed on a Rotor-Gene Q thermocycler (Qiagen Inc., Hilden, Germany) [22].

## Sensitivity, efficiency and copy number calculations

To assess CDV test sensitivity and efficiency for samples tested in NY, a 1:10 dilution series of a synthesized plasmid control containing the primer and probe binding sites and known copy number was used to generate a standard curve from 50 copies to 50,000 copies. The standard curve generated from the CDV positive control plasmid was also used to calculate the total CDV copy number in the RNA extracts from the vaccine control and clinical samples using the following equation solving for x (copy number): $x = e^{((ABS(y-b)/(m))}$. Since we used 2 μL of template in each PCR reaction, we then divided the calculated copy number (x value) by 2 (copy number represented in 1 μL), and that number was then multiplied by the total RNA extract volume (amount of the final elution volume of each RNA extraction kit). Data was normalized in this manner to account for different elution volumes for each RNA extraction kit. This allowed normalization and comparison of total CDV viral copy recovery in the total RNA extract across platforms. The calculated copy numbers/sample from replicates were then averaged and graphed with standard deviations. PCR efficiency was calculated using the equation: $e = (10^{(-1/slope)}-1)*100)$. Two-tailed T-tests were used to determine significant differences between the Biomeme POC and standard laboratory platform performance.

## Animal samples

**RNA*later*™ preserved wild animal samples.** Animal samples preserved in RNA*later*™ collected from animals that died naturally during two CDV disease outbreaks in wild mesocarnivores in Austria (2011–2013, 2018) were imported from the Research Institute of Wildlife Ecology, University of Veterinary Medicine, Vienna, Austria to the Bronx Zoo, New York, USA in 2018. This included a total of 26 samples from 3 animals, a pine martin (*Martes martes*), red fox (*Vulpes vulpes*) and a European badger (*Meles meles*). Sample types included heart, liver, kidney, spleen, lung, lymph node, brain, and muscle. Small intestine was available from two animals (pine marten and European badger). After thawing, the surface of the

RNA*later*™ preserved tissues was swabbed in duplicate using a fine-tipped sterile swab (MW&E, NC, USA) prior to RNA extraction using the Biomeme or standard laboratory methods (Wildlife Conservation Society, Bronx Zoo, NY).

**Fresh frozen animal samples: Wild raccoons, New York, USA.** Tissues, nasal swabs and/ or hair samples were collected from raccoon (*Procyon lotor*) carcasses that originated from Central Park, New York City (NYC), USA. The raccoons were either found dead or were euthanized due to clinical signs consistent with CDV during a known, laboratory confirmed CDV outbreak in 2018, affecting approximately 175 raccoons. The animals were collected and transported by staff with NYC Parks and Recreation to the city's municipal animal shelter, Animal Care Centers of New York and in all but four cases, described below, no additional pathological examination was performed. Samples collected varied by animal and included plucked hair with hair root; swabs from foot pad, cerebellum, cerebrum, and lung; and fresh-frozen tissues from foot pad, cerebellum, cerebrum, and lung. Complete necropsy examination and a full set of tissues in 10% neutral buffered formalin was collected from four raccoons that died or were euthanized during the outbreak, and histopathology examination confirmed CDV disease in these animals. Nasal swabs were collected from 3 of the 4 animals. Additionally, hair with root were collected from an additional 48 CDV-suspect animals that were trapped and euthanized during the outbreak. Nasal swabs, fresh tissues and hair samples were archived frozen (-80˚C) prior to PCR testing (Wildlife Conservation Society, Bronx Zoo, NY, USA). After thawing, fresh frozen tissue was swabbed using a fine-tipped sterile swab (MW&E, NC, USA) prior to RNA extraction.

**Fresh frozen animal samples: Austrian wildlife, Vienna, Austria.** Animal samples were obtained from two naturally occurring CDV outbreaks in wild mesocarnivores in Austria (2011– 2013, and 2018). Samples were collected from 10 animals that died naturally and included: pine martin (*Martes martes*; n = 1), beech martin (*Martes foina*, n = 1), Eurasian otter (*Lutra lutra*; n = 1), European badger (*Meles meles*; n = 3), and red fox (*Vulpes vulpes*; n = 4) (S1 Table). Samples were archived frozen (-80˚C) until testing (Research Institute of Wildlife Ecology, University of Veterinary Medicine, Vienna, Austria), but no additional pathological examination was performed. Sample types included lung, brain, kidney, liver and heart (S1 Table). After thawing, fresh frozen tissue was swabbed using a fine-tipped sterile swab prior to RNA extraction (MW&E, NC, USA) for all Biomeme extractions conducted in New York and Vienna, and Qiagen RNA extractions conducted in New York. For samples tested using Qiagen methods in Vienna, tissues were sampled whole and made into an organ suspension prior to RNA extraction as described above.

**Histology and immunohistochemistry.** Routine processing, sectioning (5 μm), and hematoxylin and eosin staining of formalin fixed tissues from four raccoons (see above) was performed and stained tissues were histologically reviewed by a certified veterinary pathologist. Immunohistochemical (IHC) labeling for canine distemper virus antigen was performed using a primary monoclonal IgG1 anti-CDV surface envelope antibody and positive and negative controls as previously described [23].

## Results

Canine distemper virus detection was compared between Biomeme's portable POC platform (Fig 1, top) and standard laboratory methods using previously published CDV-specific primers and probe [21,22]. Comparison and validation were performed in an iterative manner, testing each step of the Biomeme POC platform independently against standard methods, and then combining each POC step sequentially (Fig 1, bottom). This included: 1. comparison between the Biomeme M1 Sample Prep Kit™ and the Qiagen QIAamp® Viral RNA minikit for

RNA extraction and CDV detection in a dilution series of DA2PP vaccine, 2. comparison of the Biomeme LyoRNA™ RT-PCR mastermix and the Qiagen QuantiFast Pathogen RT-PCR mastermix for CDV detection using a synthetic CDV plasmid positive control, 3. comparison of the Biomeme two3™ and Bio-Rad Mini-Opticon qPCR thermocycler for CDV detection with synthetic plasmid positive control, 4. comparison of RNA extraction between the Biomeme M1 Sample Prep Kit™ and the Qiagen QiAMP Viral RNA minikit for CDV detection with CDV-suspect, wild animal samples preserved in RNA*later*™, 5. comparison of the Biomeme M1 Sample Prep Kit™ in combination with the Biomeme LyoRNA™ RT-PCR mastermix to the Qiagen kits (RNA extraction and mastermix) for CDV detection from fresh tissues and nasal swabs from CDV-suspect, wild raccoons, and finally, 6. comparison of test performance and results for CDV detection in samples from CDV-suspect, wild animals using the complete Biomeme platform (M1 Sample Prep Kit™, CDV Go-Strips™, and Biomeme two3™ thermocycler) or standard laboratory equipment and methods at an independent virology laboratory in Austria.

## Comparison of viral RNA recovery using the Biomeme M1 Sample Prep Kit™ or QIAamp® Viral RNA extraction kit

To compare viral RNA recovery between the Biomeme POC M1 Sample Prep Kit™ and the QIAamp® Viral RNA extraction kit, a dilution series of a reconstituted, commercially available modified live DA2PP vaccine was created and extractions were performed using both kits. RT-qPCR was performed using the QuantiFast Pathogen RT-PCR + IC Kit reagents (Qiagen) and our standard Bio-Rad MiniOpticon™ real-time thermocycler (standard lab methods) in singlicate, and extraction kit performance was based on the number of detected viral RNA copies averaged over three independent experiments and plotted with standard deviations (Fig 2). Copy number recovery was determined from a standard curve of a dilution series of plasmid positive control. No statistical difference in viral copy number recovery was detected between the two extraction methods at any of the dilutions using two-tailed student T-tests (Fig 2).

## Comparison of CDV detection using Biomeme LyoRNA™ RT-PCR or QuantiFast Pathogen RT-PCR mastermix

Sensitivity of CDV detection using either the Biomeme LyoRNA™ RT-PCR mastermix or the QuantiFast Pathogen mastermix in combination with the Bio-Rad MiniOpticon™ qPCR thermocycler was compared. CDV positive control plasmid was serially diluted 1:10 (50 copies to 500,000 copies) and tested in duplicate with each mastermix. Cycle threshold (Ct) values were averaged and plotted with standard deviations (Fig 3). Both mastermixes performed similarly in the dilution series and detected as few as 50 copies of CDV P gene target with no statistically significant differences between kits at any dilution using a two-tailed student t-test.

## Comparison of CDV detection using the Biomeme two3™ or Bio-Rad MiniOpticon™ thermocycler

Sensitivity and efficiency of CDV detection were compared between the Biomeme two3™ and the Bio-Rad MiniOpticon™ qPCR thermocyclers. Samples were tested in singlicate using a CDV plasmid positive control and Biomeme LyoRNA™ RT-PCR mastermix, primers and probe. Cycle threshold (Ct) values from three independent experiments were averaged and plotted with standard deviations. When compared to the Bio-Rad MiniOpticon, the sensitivity of the Biomeme two3™ was similar with detection of as few as 50 copies of the CDV target per PCR reaction. Ct value differences, ranging between 0.9 and 2.6 cycles, were slightly lower on the Biomeme two3™ thermocycler as compared to the Bio-Rad MiniOpticon™ thermocycler

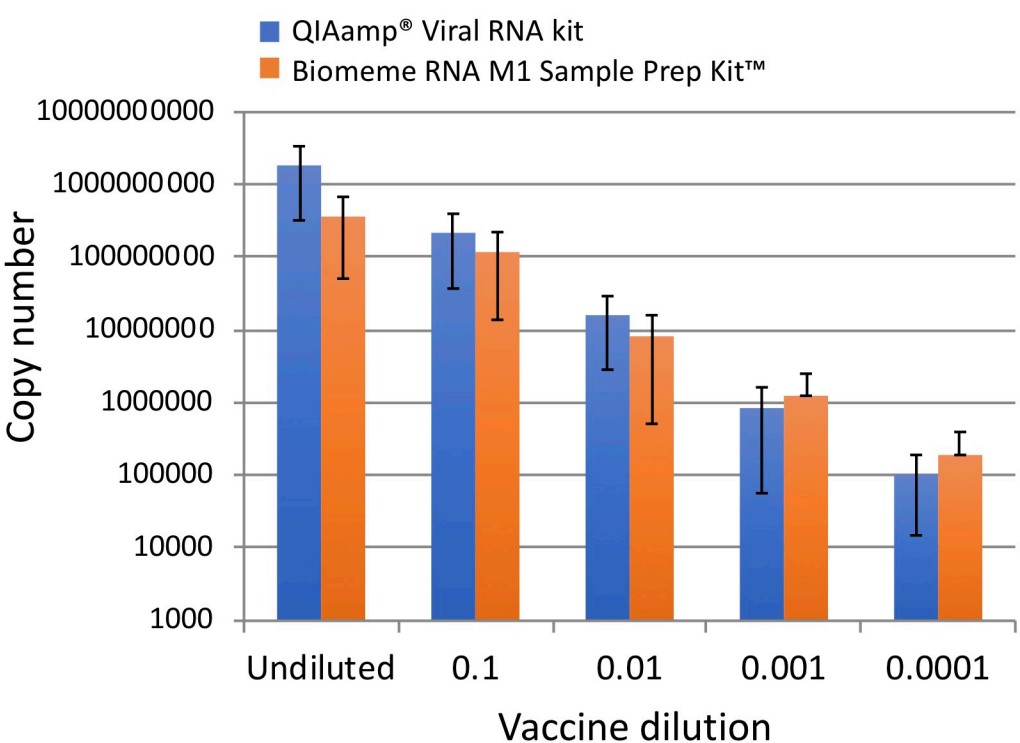

**Fig 2. Comparison of CDV RNA copy number using Biomeme or Qiagen RNA extraction kits.** Canine distemper virus copy number recovery from a ten-fold dilution series (undiluted, 1:10, 1:100, 1:1000, 1:10000) of a reconstituted modified-live, attenuated, DA2PP vaccine. Copy numbers recovered by the Biomeme RNA field extraction kit (orange bars) were compared to the Qiagen extraction kit (blue bars). Shown are standard deviations of the average from three independent experiments. There were no statistical differences (two-tailed student t-test) between the kits at any dilution (p > 0.17).

(Fig 4). On the Biomeme two3™ thermocycler, the slope of the log-linear phase of the amplification reaction was -3.16, the amplification efficiency was 107.1%, and the $R^2$ value (correlation coefficient) was 0.998 (Fig 4B). For the Bio-Rad MiniOpticon™ the slope was -3.2, the amplification efficiency was 105.4%, and $R^2$ was 0.9472 (Fig 4C).

## Comparison of RNA extraction from RNA*later*™ preserved tissues from CDV-positive wildlife: Biomeme M1 Sample Prep Kit™ or Qiagen Viral RNA extraction kit

Viral RNA extraction from swabs of RNA*later*™ preserved, known CDV-positive tissue samples from Austrian wildlife that died during a wild mesocarnivore CDV outbreak was compared using the Biomeme M1 Sample Prep Kit™ or QIAamp® Viral RNA extraction kit. RNA was extracted from swabs from heart, liver, kidney, spleen, lung, lymph node, brain, muscle and small intestine from a pine martin, red fox and European badger. RT-qPCR was performed in singlicate and all tests were performed using Biomeme LyoRNA™ RT-PCR reagent, CDV primers and probe, on the Bio-Rad MiniOpticon. Ct values were averaged across triplicate samples and plotted with standard deviations (Fig 5A). Positive CDV detection was seen in all tissues with both extraction methods. Overall, differences in Ct values between the extraction kits were seen within and between tissue types, and Ct values were consistently higher with the Biomeme extraction method in all tissue types. Ct value differences ranged between 1.9 (liver) to 8.6 cycles (brain) between the Biomeme extraction method and Qiagen

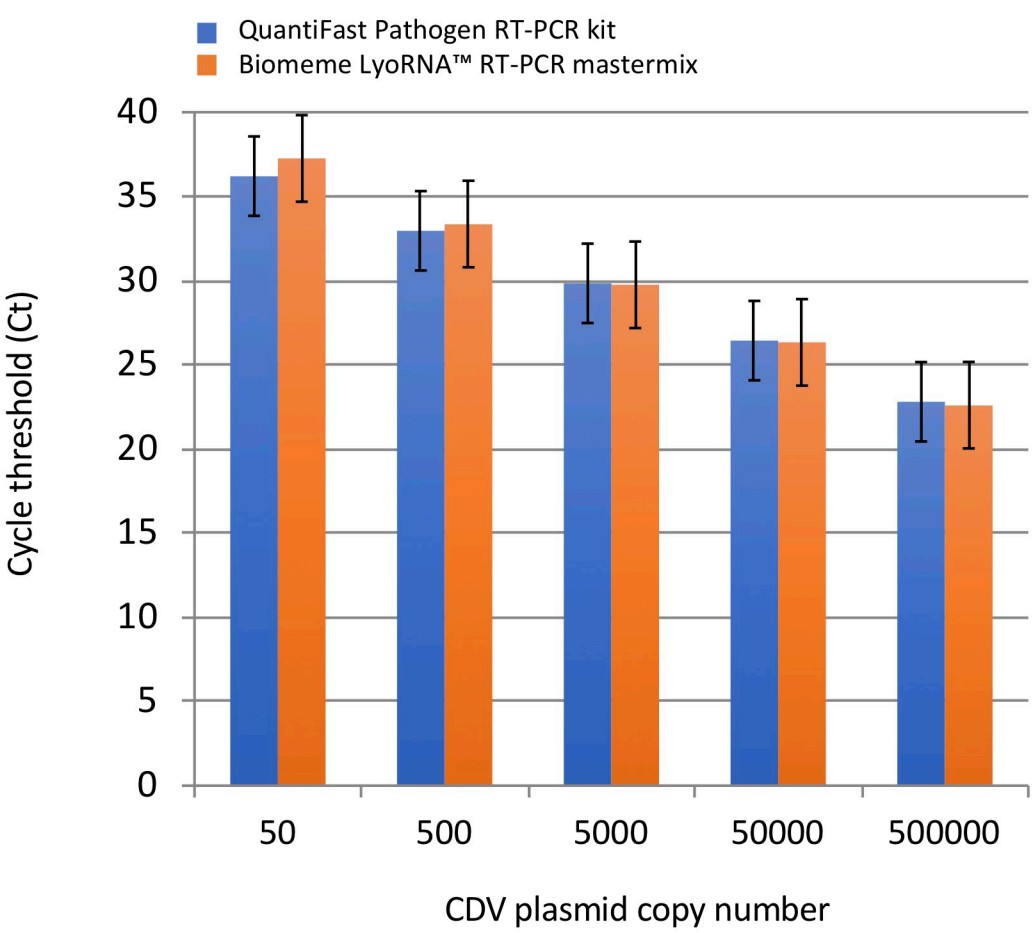

**Fig 3. Comparison of Biomeme LyoRNA™ mastermix to QuantiFast Pathogen master mix for CDV detection.** Cycle threshold (Ct) values of a standard dilution series of canine distemper virus positive control plasmid. No statistically significant differences were detected (two-tailed student t-test) in any of the duplicate samples in a comparison of Biomeme LyoRNA™ RT-PCR to QuantiFast Pathogen RT-PCR mastermixes (p > 0.08). All RT-qPCR tests were run on a Bio-rad MiniOpticon qPCR thermocycler.

(Fig 5B). The total average Ct value difference across tissues types was 5.5 cycles when using the Biomeme M1 Sample Prep Kit™ versus the QIAamp® Viral RNA extraction kit. Copy number recovery was calculated and normalized to total RNA extract volume to compensate for different elution volumes between kits. In heart, spleen, lung and brain, RNA viral copy number recovery was lower (0.5–2 logs) with the Biomeme M1 Sample Prep Kit™, but did not reach statistical significance, and was comparable in other tissues that had similar viral recovery such as liver and muscle when normalized to total viral RNA recovered in the RNA extract (Fig 5C).

## Comparison of extraction kits and reagents using fresh frozen tissues and nasal swabs from CDV-suspect animals: Biomeme's M1 Sample Prep Kit™ + LyoRNA™ mastermix or the QIAamp® Viral RNA extraction kit + QuantiFast Pathogen mastermix

Use of the Biomeme M1 Sample Prep Kit™ in combination with the Biomeme LyoRNA™ mastermix was compared to Qiagen RNA extraction and QuantiFast Pathogen mastermix in fresh

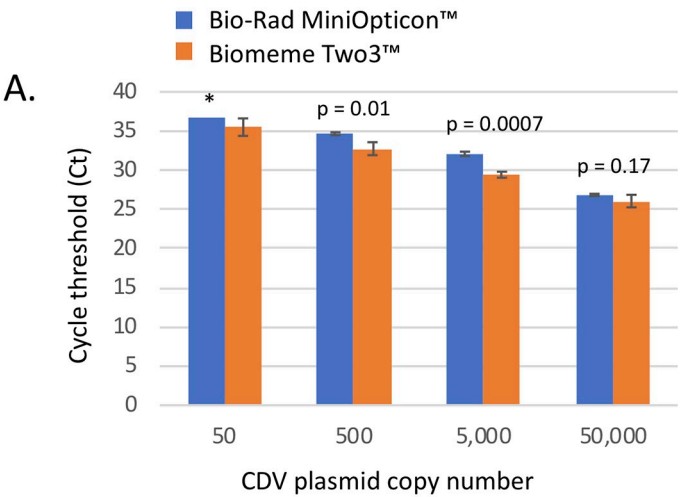

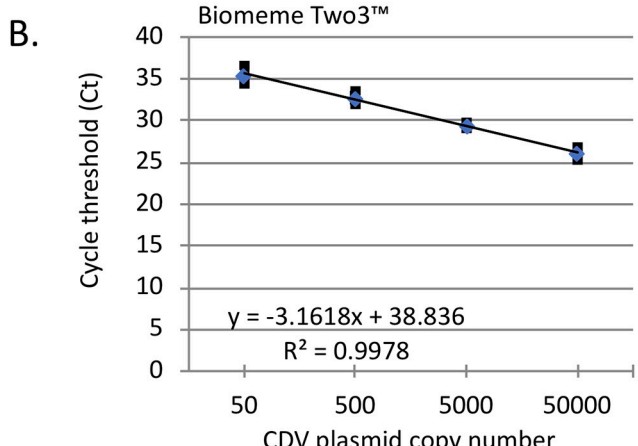

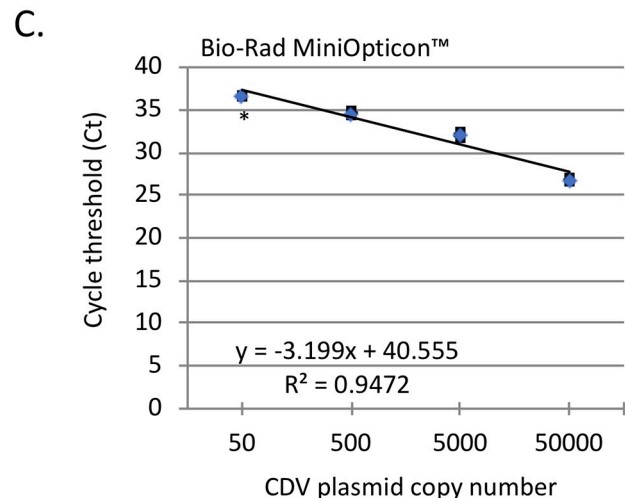

**Fig 4. Comparison of the Biomeme Two3™ and Bio-Rad MiniOpticon assay sensitivity.** A. Triplicate experiments showing cycle threshold (Ct) values and standard deviations in a ten-fold, serial dilution series of a CDV control plasmid with known copy number using Biomeme LyoRNA™ RT-PCR mastermix on the Biomeme two3™ (orange bars) or Bio-rad MiniOpticon™ (blue bars) thermocyclers. P-values are shown in samples evaluated by a two-tailed student t-test. B and C. Standard curves of a standard dilution series of a CDV control plasmid with known copy number from triplicate experiments on the Biomeme two3™ (B) or the BioRad MiniOpticon (C). (* indicates that only 1 of 3 samples tested was positive).

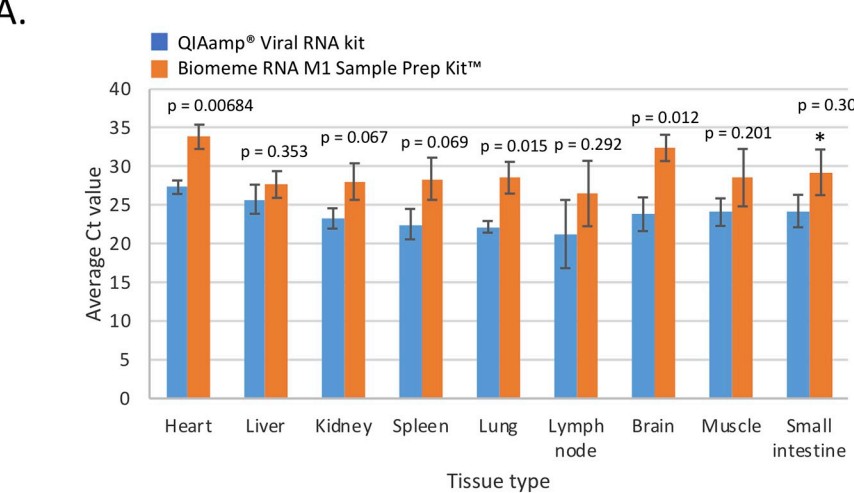

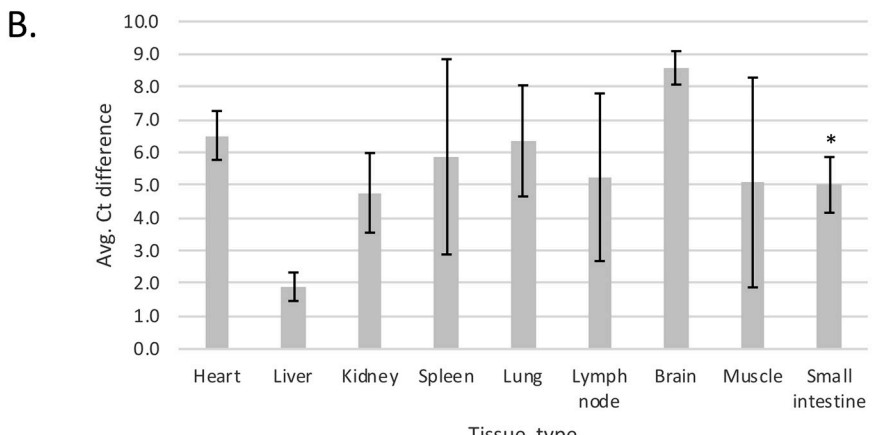

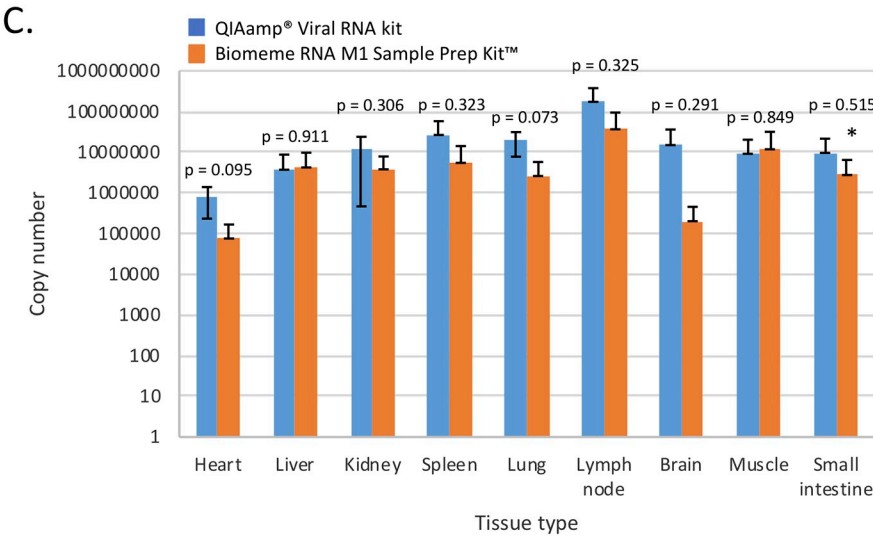

**Fig 5. Comparison of viral recovery in various tissue types stored in RNA*later*™ from a CDV outbreak in Vienna, Austria.** Data represents the average of testing a set of tissues from 3 separate animals (n = 3): a CDV positive pine martin, red fox and European badger. Standard deviations are shown, except in small intestine (* indicates tissue where only two cases were available for analysis). A. Comparison of Ct values between the Biomeme M1 Sample Prep Kit™ and QIAamp® Viral RNA extraction kit from tissue swabs. P-values are shown in samples evaluated by a two-

tailed student t-test. B. Average Ct differences between Biomeme and Qiagen RNA extraction methods for each tissue swab type (in A). C. Average calculated copy number recovery in total RNA extracts. P-values are shown in samples evaluated by a two-tailed student t-test (p > 0.07).

frozen tissue, swab, and hair samples from CDV-suspect animals collected during a CDV outbreak in raccoons (Central Park, New York City, NY; 2018). Duplicate nasal, foot pad, brain and lung swabs as well as fresh frozen brain, lung, non-haired footpad, and hair with root tissue were tested. All RT-qPCR was run on the Bio-Rad MiniOpticon™. Ct values for individual samples were averaged across the quadruplicate tests (n = 4 animals) and plotted with standard deviations (Fig 6A). All samples were positive using both platforms, and Ct values were consistently higher in all tissue types processed with the Biomeme extraction method and reagents. The Ct difference within tissue types ranged from 2.6 (hair) to 7.6 Ct difference (cerebrum). The average Ct value difference was 5.1 across samples types (Fig 6A). When normalized to account for different elution volumes, recovered viral copy number using the Biomeme platform was similar to the Qiagen platform in hair, brain (cerebrum) swab and frozen brain (cerebellum), and was 0.5–2 logs lower in nasal swab, foot pad, brain (cerebellum) swab, lung swabs, and frozen lung, foot pad, brain (cerebrum) and lung (Fig 6B), however these differences did not reach statistical significance.

## CDV immunohistochemistry of skin and hair bulb from CDV-positive raccoon samples

CDV is an epitheliotropic virus, and though affected animals can develop skin lesions (hyperkeratosis, hard pad disease), hair has not been historically used as a standard diagnostic sample. Hair can be an accessible sample in cases where internal organ samples cannot safely be collected in a way to limit human exposure, and CDV detection from hair can provide timely results to differentiate from rabies virus as a cause of infection for animals showing signs of neurologic illness. Given the challenges in clinically differentiating the two diseases, testing of hair samples could a valuable alternative approach that can decrease risk of human exposure to rabies when testing for CDV. We used immunohistochemistry (IHC) to verify that CDV virus is found in the skin and hair follicular epithelium/root bulb of infected animals, and as an additional internal positive control for our PCR results.

IHC labeling for CDV and bright-field microscopy was performed on all of the tissues that were examined histologically (by hematoxylin and eosin) from four raccoons that died or were euthanized during the 2018 CDV outbreak in New York. Positive labeling by IHC was seen in brain (4 of 4), spinal cord (1 of 1), lung (3 of 4), kidney (4 of 4), urinary bladder (2 of 4), haired skin (2 of 4) (Fig 7A and 7B), footpad (non-haired skin) (3 of 4) (Fig 7C and 7D), spleen (2 of 3), stomach (1 of 2), small intestine (1 of 2), large intestine (2 of 2), and pancreas (1 of 1); no labeling was seen in heart (n = 4) or liver (n = 4).

## Comparison of extraction kits for screening hair root bulb samples from CDV infected animals: Biomeme's M1 Sample Prep Kit™ + LyoRNA™ mastermix or the QIAamp® Viral RNA extraction kit + QuantiFast Pathogen mastermix

To further test the Biomeme POC platform for a non-invasive sample type, CDV screening of duplicate hair bulb (with root) samples from a total of 52 dead or euthanized raccoons from the 2018 Central Park outbreak were collected from the dorsal surface of a forefoot. Tests were performed as above to compare the Biomeme platform (Biomeme M1 Sample Prep Kit™ and

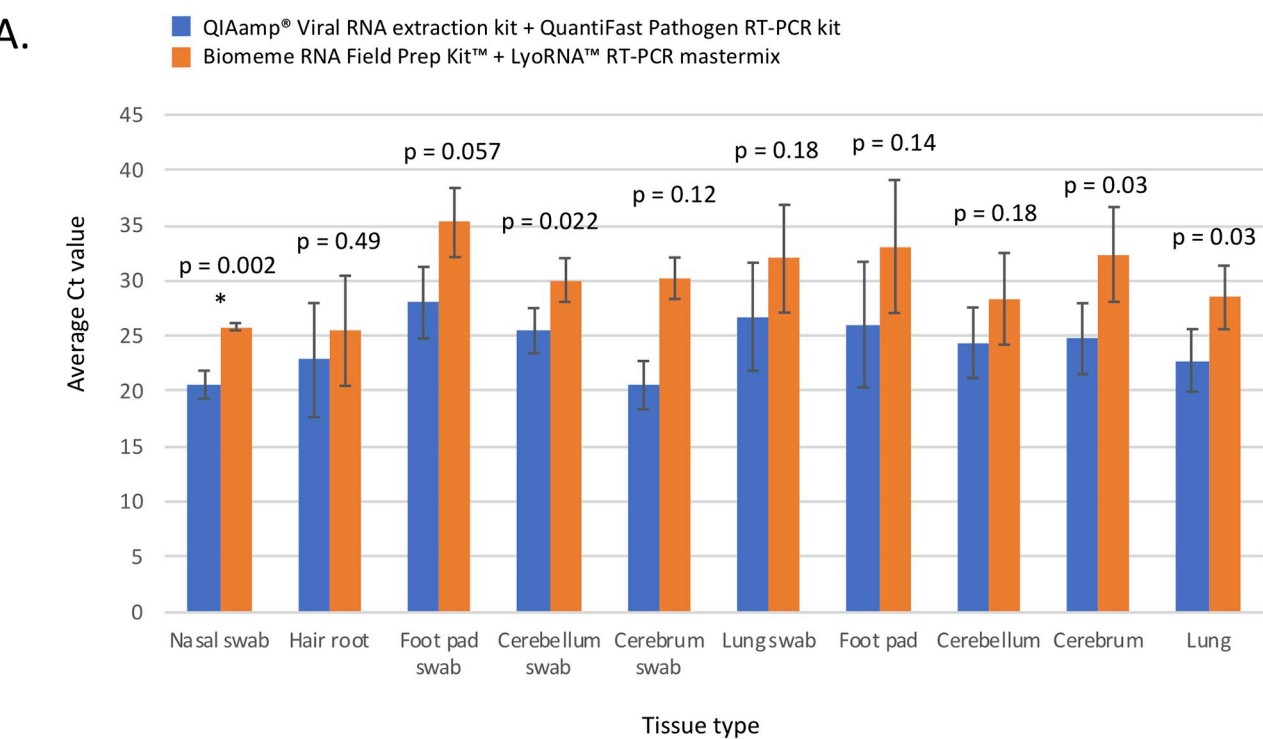

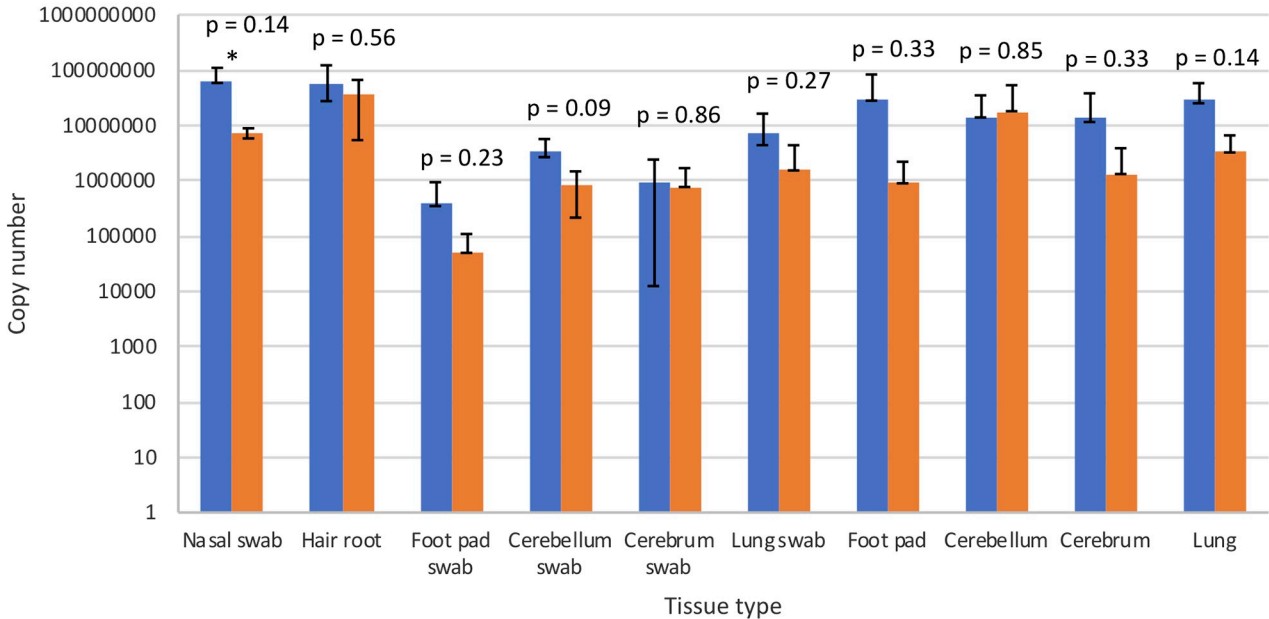

**Fig 6. Comparison of CDV detection in various sample types from a CDV outbreak in New York, NY: Biomeme M1 Sample Prep Kit™ combined with their LyoRNA™ mastermix versus Qiagen RNA extraction and QuantiFast Pathogen mastermix.** A. Comparison of the average Ct values in fresh frozen tissues, swabs and hair from four CDV-suspect raccoons. B. Average copy number recovery in total RNA extracts from the four cases. In both graphs, the data represent the average of quadruplicate testing, and standard deviations are shown. (* indicates the samples available from three rather than four raccoons (nasal swab)). P-values are shown in samples evaluated by a two-tailed student t-test.

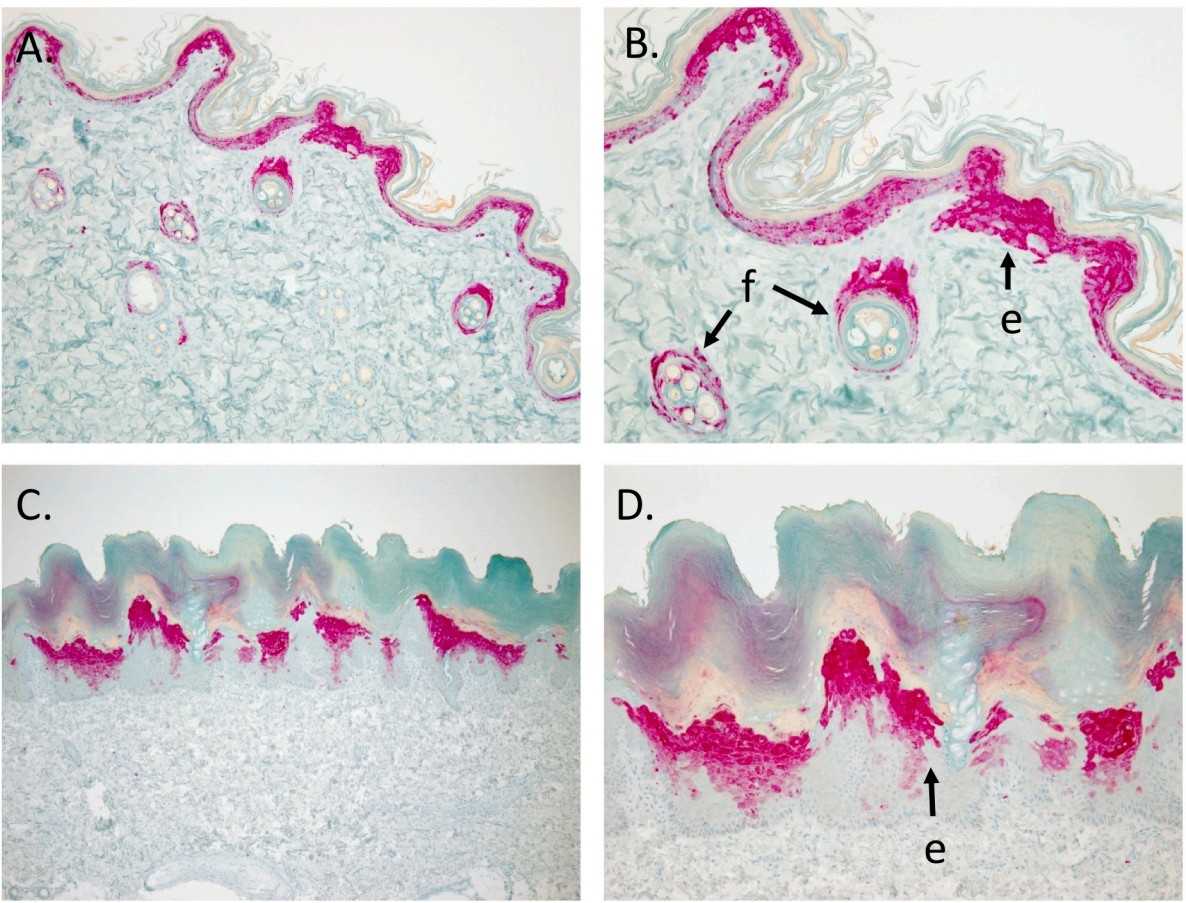

**Fig 7. CDV immunohistochemistry staining of haired skin and footpad from a CDV-positive raccoon.** Immunohistochemical labeling with a monoclonal IgG primary antibody to CDV viral envelope protein antigen (fast-red staining) in haired skin (A and B) and non-haired footpad (C and D) from a CDV positive raccoon. Cytoplasmic immunohistochemical labeling is present in epidermal (e) and follicular epithelial cells (f) in haired skin (A. 200X; B. 400X) and epidermal epithelial cells (e) in non-haired footpad (C, 100X; D, 200X).

LyoRNA™ mastermix) to our standard laboratory protocol (QIAamp® Viral RNA extraction kit in combination with the QuantiFast Pathogen RT-PCR +IC Kit, Qiagen). All RT-qPCR was run on the Bio-Rad MiniOpticon™ in singlicate with positive and negative controls. All hair samples were positive for CDV regardless of platform used, and negative controls were negative on both platforms (Fig 8A). The range of Ct value differences in individual animals between the Biomeme platform and Qiagen was 0.2–8.6 cycles. The average Ct value difference across all 52 samples was 3.61 cycles, with a standard deviation of 2.01 cycles, which was statistically significant (P < 0.001) (Fig 8B). However, when we adjusted for total CDV copy number recovery in the RNA extract, we observed no statistically significant difference in CDV detection (P = 0.76) (Fig 8C).

### Independent laboratory validation: Comparison of the Biomeme POC platform using lyophilized reagents, primers, and probe to standard virology laboratory methods in fresh frozen tissue from CDV-suspect animals

Fresh tissue samples from CDV-suspect wild mesocarnivores were collected and archived at -80C during a CDV outbreak in Austria 2011–2013, and in the spring of 2018. Duplicate samples were tested using both the Biomeme platform (Biomeme M1 Sample Prep Kit™,

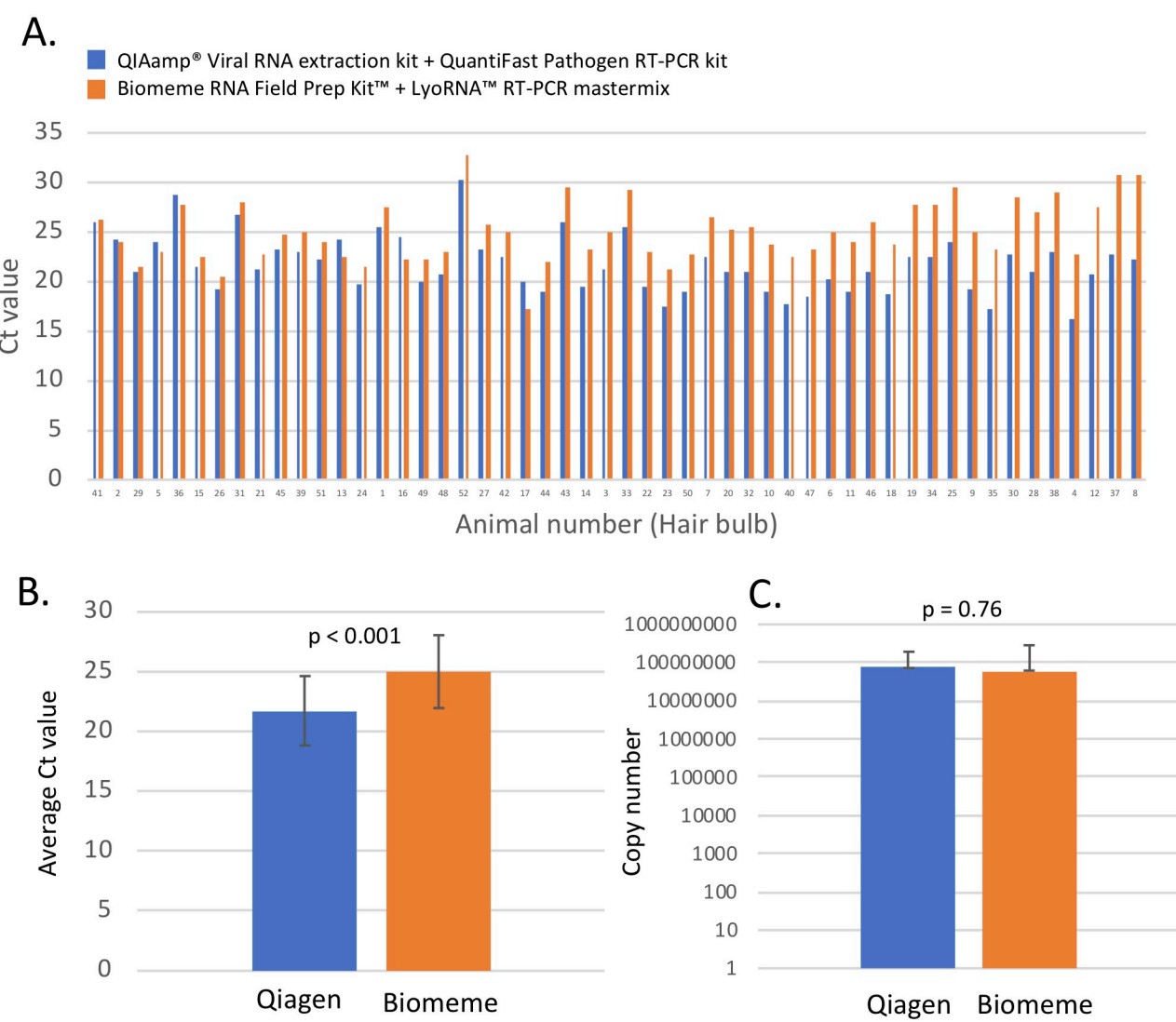

**Fig 8. Comparison of RT-qPCR CDV detection (Ct values) using Biomeme POC and standard laboratory methodology in raccoon hair samples from a CDV outbreak in New York, NY.** A. Comparison of Ct values between the two platforms in hair samples from 52 raccoons that died during a 2018 CDV outbreak. Ct values are shown for each individual animal and values are plotted in order of increasing Ct value difference. B. Average cycle threshold difference between Qiagen and Biomeme test with standard deviations are shown (p< 0.001, two-tailed student t-test). C. Average calculated copy number recovery in total RNA extracts between Qiagen and Biomeme test. There was no statistical difference (two-tailed student t-test) between the kits in CDV copy number detection of hair samples (p = 0.76).

lyophilized CDV 'Go-strips'™, and Biomeme two3™ thermocycler) and standard laboratory-based methods and protocols (Institute of Virology, University of Veterinary Medicine, Vienna, Austria). Testing was performed on brain, lung, kidney, liver, and heart from 10 animals: pine martin (n = 1), beech martin (n = 1), European badger (n = 3), Eurasian otter (n = 1), and red fox (n = 4) (S1 Table). We found 100% concordance in detection between the Biomeme platform and the virology lab in CDV detection in kidney (6 of 9 positive), liver (2 of 3 positive), heart (1 of 1 positive), and lung (5 of 7 positive) samples (Table 1). We found that in brain samples, 6 of 6 were positive with the Biomeme platform while 5 of 6 were positive with the virology laboratory testing methods. Upon reviewing the Biomeme data from the positive *Lutra lutra* brain sample 2054, we observed an attenuated amplification curve in the

**Table 1. Results of CDV RT-qPCR testing using the Biomeme POC platform or Virology Laboratory methods in fresh frozen tissue samples from CDV suspect, Austrian mesocarnivores.**

| Tissue type | Number of Samples | Biomeme POC Platform | | Virology Lab Methods | |
|---|---|---|---|---|---|
| | | Positive | Negative | Positive | Negative |
| Lung | 7 | 5 | 2 | 5 | 2 |
| Brain | 6 | 6 (*5) | 0 (*1) | 5 (*5) | 1 (*1) |
| Kidney | 9 | 6 | 3 | 6 | 3 |
| Liver | 3 | 2 | 1 | 2 | 1 |
| Heart | 1 | 1 | 0 | 1 | 0 |
| **Total** | **26** | **20 (*19)** | **6 (*7)** | **19 (*19)** | **7 (*7)** |

* Denotes final data after brain samples from 2054 was retested using both platforms.

positive result, amplifying and plateauing within 5 cycles. This sample was retested and confirmed to be negative by the Virology lab, and also negative using the Biomeme platform. Therefore we conclude that the initial test result was a false positive. Overall, 96.1% (25 of 26 samples) concordance was observed between platforms initially, and 100% concordance was observed after retest from suspect CDV cases.

## Discussion

This study was designed to test the performance of an innovative point-of-care qRT-PCR platform to rapidly and accurately diagnose canine distemper virus (CDV) in field or laboratory settings. The Biomeme platform consists of rapid cellular lysis (10 minutes) and RNA extraction (3–5 minutes) steps, pre-packaged 'Go-strips'™, and the hand-held, Biomeme two3™ thermocycler equipped with a smartphone for fluorescent detection and data collection (78 min, 45 cycles) (Biomeme Inc. Philadelphia, PA, USA). This portable qPCR platform can be run off-grid with a battery or solar power source. Color-coding of reagents in the extraction process easily overcomes training and language barriers. Shelf-stable, lyophilized reagents can be combined with target-specific primers and probes that are incorporated into a bead in pre-packaged PCR reaction tubes, so called 'Go-Strips'™. Reconstitution of the beads with DNA or RNA template prepares the sample for PCR. Results can be downloaded to the phone, laptop computer or uploaded for cloud-based storage and retrieval.

Our comparisons of this platform to those of two different standard laboratory-based methodologies show that overall, CDV detection using the Biomeme platform was similar to standard laboratory methodologies for plasmid positive control and vaccine control testing. In addition, sensitivity (to 50 copies per PCR reaction) and efficiency (90–110%) were consistent between platforms with positive control synthetic plasmid, and the latter was well within acceptable limits for traditional laboratory methods [21,24]. When compared across different tissue types from different species, and under different storage conditions (RNA*later*™, frozen), the Biomeme platform consistently detected CDV positive samples; however, the Ct values were generally higher than with traditional Qiagen RNA extracted samples. Several of the experiments with higher Ct values were conducted while examining only one variable, which allowed us to identify the Biomeme RNA extraction step as the likely cause for this difference in our tests. Further work is needed to determine if the reduction in viral detection is due to either reduced binding of all available RNA to the column or less efficient release of RNA off the column, and if any optimization can be done to improve recovery. We found that RNA extraction from swabs of tissue and whole tissue samples had a lower RNA recovery (0.5–2 logs lower) with the Biomeme methodology than with Qiagen extraction when normalized for total CDV viral RNA in the extract,

though these results did not reach statistical significance. So, while overall CDV detection with both platforms was consistent, (26 of 26 after retest), the lower observed recovery of viral RNA with the Biomeme extraction methods, especially in samples with low viral load, has the potential to result in false negative results. Further testing with a variety of different viral titers will be beneficial in ascertaining how this kit performs in a variety of clinical CDV samples that contain low versus high viral loads. Nevertheless, given the portability of the platform, ease of use in the field, and consistent detection of CDV from known and suspect positive samples across tissue and sample types in our study, this platform is a useful point-of care testing tool for detecting CDV.

RNA*later*™ is commonly used in the field as a method of preserving nucleic acids in tissue or swab samples from wildlife. We therefore tested the performance of the Biomeme platform with RNA*later*™ preserved tissues from known CDV positive cases. No observable differences in CDV viral recovery were seen in a comparison using swabs of fresh frozen or RNA*later*™ preserved tissues across platforms. Provided the samples are stored optimally according to the manufacturer's specifications for RNA preservation, RNA*later*™ is a suitable preservation method for samples and compatible using the Biomeme POC platform. However, given our small sample size, future comparisons between samples collected from animals with known high or low viral loads, or short and long term, ambient vs frozen RNA*later*™ archiving to test RNA preservation and degradation would be especially valuable to establish minimum detection thresholds for CDV using the Biomeme platform.

In addition to positive results in classic tissues targeted for CDV testing such as frozen brain and lung, our data show that hair bulb was a reliable, non-invasive sample for CDV detection in virus positive animals. IHC staining of CDV in the epidermal and follicular epithelial cells was consistent with the PCR results. Hair sample collection is easy, can be performed appropriately with minimal training, and can be opportunistically performed in live or dead animals making it a useful sample to collect from wildlife in the field or in a clinical setting. Non-invasive sampling reduces risk associated with handling sick or dead carnivores, especially those that could also be infected with rabies. Rabies is a zoonotic disease that can present with similar clinical neurological symptoms to CDV, and co-infection with both can occur [25]. There are currently no POC tests for rabies. However, a CDV negative POC test result from a non-invasively collected sample would serve to raise a higher suspicion of rabies or other neurologic disease processes for field personnel. In our testing, we found 100% concurrence in our results comparing the Biomeme and standard laboratory platforms. However, as mentioned above, higher Ct values were seen with the Biomeme platform (3.61 cycles, with a standard deviation of 2.01 cycles, n = 52 samples). When compared to other tissues types, the Ct differences across the hair samples were comparable, suggesting hair as a good, non-invasive sample for CDV POC testing. All raccoons tested in our study were found dead or euthanized after exhibiting clinical symptoms, so additional testing of hair samples comparing early, mid and late stage CDV infection will be especially valuable for determining if hair bulb is more or less valuable at certain stages of the disease.

Over the past several years, a variety of methods have become available for CDV detection, with a range of applications and sensitivities. New developments for rapid, quantitative, reverse transcription, recombinase polymerase amplification (RT-RPA) and isothermal polymerase gene amplification (RT-iiPCR) techniques for CDV detection have been recently described [26,27]. Field-capable equipment using these techniques is heavy (1.75 kg for a Genie III used in RT-RPA detection, and 2.1 kg for a POCKIT™ iiPCR analyzer) and requires standard Qiagen commercial kits for RNA extraction. The latter require additional equipment and more reagents and time than the Biomeme M1 Sample Prep Kit™, which can be used off-grid with solar power and without equipment such as centrifuges and heat blocks [14,15]. The entire Biomeme POC platform is lightweight and very portable, and all of the equipment and supplies are able to fit in

a standard sized backpack. The two3™ thermocycler plus case is 28 cm x 25.4 cm x 12.7 cm, and at 0.5 kg, making it is considerably lighter than the Genie III or POCKIT™ iiPCR analyzer. In our experience, the Biomeme equipment is also easier to use than other platforms, and, like their extraction kit, the Biomeme thermocycler can be used in remote field settings that lack electricity. When lyophilized, the reagents (lyophilized Go-strips™) are also shelf stable and do not require cold storage. Biomeme also offers an option to include inhibition control lyophilized Go-strips™ in cases where PCR inhibition may be problematic. In our field-tests using the Biomeme two3™ qPCR thermocycler, when fully charged, we have successfully performed up to five consecutive thermocycle runs on a single charge, and the unit is compatible with and able to be charged by a solar-charged battery such as the Goal 0™ (Goal Zero, Bluffdale, UT) (Seimon and Brown, unpublished results).

One notable limitation of the Biomeme two3™ thermocycler is that the unit has only three sample wells. So, if both a positive and negative control are run, the user is limited to one unknown sample for each PCR run. This is less problematic when dealing with individual animal POC testing or in projects with fewer samples or low throughput. Additionally, Biomeme has recently released a three9™ device, a slightly larger platform with a total of 9 reaction wells with three color channels, which can test up to 27 targets.

CDV affects a wide variety of carnivore species, including mustelids, procyonids, ursids, canids (domestic and wild), felids and marine mammals [2,28–30]. It is considered among the most widespread multi-host pathogens [31]. CDV is a recognized conservation threat to a number of endangered carnivores. Our understanding of CDV ecology, disease transmission and infection outcomes is far from complete and advancing our knowledge is critical for mitigating its effects in wildlife. For example, CDV has become a major conservation threat to endangered populations of Amur tigers (*Panthera tigris altaica*) and Far Eastern leopards (*Panthera pardus orientalis*) in the Russia Far East [5,7,8]. This is in addition to the threats faced from poaching, habitat loss and tiger human conflict [32,33]. Serological, genetic and demographic studies in domestic dogs and mesocarnivores are needed to determine the role that wildlife and domestic dogs play for both maintaining CDV in the environment and as a source of infection for tigers [6,7]. Having a mechanism for rapid, effective, and field-friendly CDV diagnosis will improve conservation efforts to track disease spread and identify reservoir species, which will in turn lead to better methods for monitoring vulnerable populations and development of solutions to support the conservation of endangered carnivores in the wild. Additionally, adaptation of this platform to test for other pathogens will expand the utility of POC for health monitoring and disease surveillance in conservation, especially in remote, under-resourced regions, and can be a valuable tool in rescue and rehabilitation centers, shelters, and boarding facilities for wildlife and domestic animals as well.

## Supporting information

**S1 Table. Individual sample results and metadata from CDV suspect, Austrian mesocarnivores.** qPCR results from 26 samples of CDV suspect animals comparing standard laboratory practices at the Virology Laboratory (University of Veterinary Medicine, Vienna, Austria) to the Biomeme POC platform.
(DOCX)

## Acknowledgments

We are grateful to Martin Gilbert, Nadezhda Sulikhan and Dale G. Miquelle for their early feedback and support during the development of this project, and Martin Gilbert for

comments and feedback on this manuscript. Biomeme Inc. was not involved in this study. All reagents and equipment using this platform were purchased from Biomeme Inc. for use in this study.

## Author Contributions

**Conceptualization:** Denise McAloose, Paul P. Calle, Chris Walzer, Tracie A. Seimon.

**Data curation:** Ania Tomaszewicz Brown, Denise McAloose, Sally Slavinski, Robin Brennan, Tracie A. Seimon.

**Formal analysis:** Ania Tomaszewicz Brown, Denise McAloose, Angelika Auer, Annika Posautz, Tracie A. Seimon.

**Investigation:** Ania Tomaszewicz Brown, Denise McAloose, Angelika Auer, Annika Posautz, Sally Slavinski, Robin Brennan, Chris Walzer, Tracie A. Seimon.

**Methodology:** Ania Tomaszewicz Brown, Denise McAloose, Angelika Auer, Annika Posautz, Tracie A. Seimon.

**Project administration:** Ania Tomaszewicz Brown, Sally Slavinski, Robin Brennan, Chris Walzer, Tracie A. Seimon.

**Resources:** Denise McAloose, Paul P. Calle, Angelika Auer, Annika Posautz, Sally Slavinski, Robin Brennan, Chris Walzer.

**Supervision:** Denise McAloose, Paul P. Calle, Chris Walzer, Tracie A. Seimon.

**Validation:** Ania Tomaszewicz Brown, Angelika Auer, Annika Posautz, Tracie A. Seimon.

**Visualization:** Ania Tomaszewicz Brown, Tracie A. Seimon.

**Writing – original draft:** Ania Tomaszewicz Brown, Denise McAloose, Tracie A. Seimon.

**Writing – review & editing:** Ania Tomaszewicz Brown, Denise McAloose, Paul P. Calle, Angelika Auer, Annika Posautz, Sally Slavinski, Robin Brennan, Chris Walzer, Tracie A. Seimon.

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
