## [Decision Letter · Decision Letter 0]

24 Feb 2020

PONE-D-20-01230

Development and validation of a portable, point-of-care canine distemper virus qPCR test

PLOS ONE

Dear Dr Seimon

Thank you for submitting your manuscript to PLOS ONE. After careful consideration, we feel that it has merit but does not fully meet PLOS ONE’s publication criteria as it currently stands. Therefore, we invite you to submit a revised version of the manuscript that addresses the points raised during the review process.

Many thanks for submitting your manuscript to PLOS One

The manuscript was reviewed by two reviewers. Due to a difficulty in finding reviewers, I have provided one of the reviews myself as it is akin to some work which I have done

The comments are generally only minor, but if you could write a response to reviewers then it will expedite things when you resubmit.

I wish you the best of luck with your revisions

Many thanks

Simon

We would appreciate receiving your revised manuscript by Apr 09 2020 11:59PM. To enhance the reproducibility of your results, we recommend that if applicable you deposit your laboratory protocols in protocols.io, where a protocol can be assigned its own identifier (DOI) such that it can be cited independently in the future. For instructions see: http://journals.plos.org/plosone/s/submission-guidelines#loc-laboratory-protocols

We look forward to receiving your revised manuscript.

Kind regards,

Simon Russell Clegg, PhD

Academic Editor

PLOS ONE

Journal Requirements:

2. In your Methods section, please provide additional location information, including geographic coordinates for the data set (locations of animal capture), if available.

3. Please provide the method(s) of euthanasia in the Methods section of your manuscript, where applicable and known.

4. Please include a caption for Figures 1 and 8.

Reviewers' comments:

Reviewer's Responses to Questions

**Comments to the Author**

1. Is the manuscript technically sound, and do the data support the conclusions?

Reviewer #1: Yes

Reviewer #2: Yes

2. Has the statistical analysis been performed appropriately and rigorously? 

Reviewer #1: Yes

Reviewer #2: Yes

3. Have the authors made all data underlying the findings in their manuscript fully available?

Reviewer #1: Yes

Reviewer #2: Yes

4. Is the manuscript presented in an intelligible fashion and written in standard English?

Reviewer #1: Yes

Reviewer #2: Yes

5. Review Comments to the Author

Reviewer #1: Overall, this is a well-written manuscript with sound methodology and the authors have clearly conveyed the importance of the development of a field-deployable point-of-care diagnostic assay for canine distemper virus. The results show that the establishment of this CDV Biomeme assay has the potential to be a useful diagnostic tool for both wildlife biologists and in conservation efforts. The confirmation of hair as a reliable sample source for CDV diagnostics is particularly advantageous as risk is minimized to field personnel without compromising the ability to generate rapid, actionable data.

I am concerned by the disagreement between the laboratory standard assay and the Biomeme assay. The authors report that one sample derived from brain tissue was reported as positive by the Biomeme assay, but negative with the standard assay. This is inconsistent with the frequent discussion point throughout the manuscript that the Biomeme assay is less sensitive than the laboratory standard assay. The manuscript would be improved by the inclusion of a follow-up to confirm the results for the sample in question with both assays.

I believe that the study and described assay is a valuable addition to the field in its current state. As the authors note, clinical manifestations of rabies virus in wildlife may be confused for any number of other pathogens including CDV; however, rabies virus poses a high risk to field personnel. The assay could be improved in the future with the development of a multiplexed assay targeting CDV, Rabies virus, and an internal control with low variability between related species (e.g. HPRT or GAPDH) to ensure that the nucleic extraction was successful.

Reviewer #2: This is a very interesting, and well written manuscript comparing diagnostic techniques for the detection of canine distemper virus. The authors have done an excellent job (and should be highly commended), so I have very few, and only generally minor comments.

Line 35- using a positive control plasmid

Line 79- I think here it is worthwhile making a bit more of the importance of the diagnostics. As the disease clinical signs can sometimes resemble rabiles then this is an important thing to state here. Not least because diagnostics for CDV in the field will mainly be cadaver based as there is no real treatment option for CDV

Line 88- maybe have highlighted may sound better?

Line 99- four should be in words

RNA extraction using the Biome RNA field prep kit- are the reagents for the different buffers- e.g. BLB available online? If so could you put a link in to these? Or are they unknown for commercial reasons?

Line 136- a comma after expelling may help with reading flow

You have several different extraction procedures- would this not affect levels of DNA extracted and thus results?

In several places you say that the samples were tested in singlicate and then in the results you mention some in duplicate and some in triplicate. This was a bit unclear and confusing so maybe you could look at explaining this a little differently? It is also more common to do samples in duplicate or triplicate

You mention the animals which died during CDV outbreaks. Was any pathology done on these animals to confirm the disease as they could have died of anything?

Line 317- 3 should be in words

Line 326- 27- three should be in words and the n = 3 seems a bit repetitive? For the CDV detection comparison with Biome- were the results in Lines 356-360 significant?

Again, the use of triplicates when the methodology says singular is a little confusing

Line 429- four should be in words

Line 430- comma after graphs

Line 432- 3 and 4 should be in words

Line 438- the second ‘been’ can be removed

Line 448- 4 should be in words

Line 534- in my experience there is a lot of DNA and RNA lost using the Qiagen kits due to the use of the filters which tend to be relatively poor at releasing DNA after washing steps

Line 536- you say that this is likely due to the RNA extraction- which part of it, and could you optimise it in any way?

Line 543- I would always avoid saying test with larger sample sizes- because it raises the simple question ‘why didn’t you do that then?’ (although I am not that cruel). Maybe state that using a variety of different viral titres would be beneficial?

Line 564- and animals in a clinical setting

Line 573- comma after tissue types

Line 573-575- consider rewording this as two mentions of other tissue types makes it harder to follow

Line 579- comma after several years,

Line 588 and 589- these two sentences may benefit from being linked

Line 597- 5 in words

Line 600 and 603- 3 in words

Acknowledgements- can you please just confirm that Biomeme were not involved in the study and didn’t supply reagents etc?

6. PLOS authors have the option to publish the peer review history of their article (what does this mean?). If published, this will include your full peer review and any attached files.

Reviewer #1: No

Reviewer #2: No

---

## [Author Response · Author response to Decision Letter 0]

24 Mar 2020

The response to reviewer comments is also included in the new cover letter that has been uploaded, and the comments and responses by our authors are indicated in bold to make it easier to read. All lines referenced int he manuscript refer to the Revised manuscript with track changes file. 

PONE-D-20-01230

Development and validation of a portable, point-of-care canine distemper virus qPCR test

PLOS ONE

Journal Requirements:

2. In your Methods section, please provide additional location information, including geographic coordinates for the data set (locations of animal capture), if available.

Location data was not collected on animals that died naturally or were euthanized during the CDV outbreaks in Austria and Central Park, NY. The samples were opportunistically collected after the animals had been found deceased and no location information, other than that provided in the methods, were obtained because these animals were not collected for the purpose of a study.

3. Please provide the method(s) of euthanasia in the Methods section of your manuscript, where applicable and known.

We have included the following information in second sentence of the ethics statement in lines 143-144 of the method section in the revised manuscript with track changes: “Following AVMA Euthanasia Guidelines Raccoons were sedated with intramuscular ketamine prior to the IV or IP injection of sodium pentobarbital.”

4. Please include a caption for Figures 1 and 8.

The caption for Figure 1 has now been added at line 356. We also noticed that the caption and referenced for Figures 7 and 8 were mislabeled in some places, which has now been corrected in line 519 For Figure 7, and for Figure 8 (lines 549-554).

We have inserted our caption for our Supporting Information Table S1 at the very end of the manuscript in the manuscript file, and removed the caption from the Supporting Information Table S1 file.

Review Comments to the Author

Reviewer #1: 

1. I am concerned by the disagreement between the laboratory standard assay and the Biomeme assay. The authors report that one sample derived from brain tissue was reported as positive by the Biomeme assay, but negative with the standard assay. This is inconsistent with the frequent discussion point throughout the manuscript that the Biomeme assay is less sensitive than the laboratory standard assay. The manuscript would be improved by the inclusion of a follow-up to confirm the results for the sample in question with both assays.

We thank the reviewer for calling attention to this disagreement. The sample in question, brain sample from Lutra lutra 2054, was retested again at the facilities in Vienna, and was confirmed to be negative for CDV. The Vienna lab also tested the sample for PCR inhibition and two other CDV targets using two primer sets for the H gene, and it was negative for CDV with no PCR inhibition observed. Therefore we are confident that this sample is, in fact, negative. In re-reviewing the Biomeme data it appears that the initial positive result had a strange amplification curve, amplifying and plateauing within 5 cycles instead of ~15 cycles as seen with the other positive samples. The sample also had a very low Ct of 19.69, which if true would fall well within the sensitivity of the standard Virology lab assay, but the lab instead found it negative. This sample was also negative in a second qPCR test using the Biomeme platform, so we now know the initial result was a false positive. We have therefore kept the data “as is” in the manuscript, however we flagged the result as a false positive and address this disagreement in light of the new results in line 40 of the abstract, lines 585-591 in results, in Table 1 and in S1 Table in the revised manuscript with track changes. We feel it important to keep the initial result as it may be informative to researchers who encounter or identify similar findings of false positive readings when using this platform. 

We added the following to lines 39-41: “CDV detection using the Biomeme platform was similar in 25 of 26 samples from suspect CDV cases when compared to standard virology laboratory testing. One false positive was observed that was negative upon retest.”

We added the following to lines 585-591: “Upon reviewing the Biomeme data from the positive Lutra lutra brain sample 2054, we observed an attenuated amplification curve in the positive result, amplifying and plateauing within 5 cycles. This sample was retested and confirmed to be negative by the Virology lab, and negative using the Biomeme platform. Therefore we conclude that the initial test result was a false positive. Overall, 96.1% (25 of 26 samples) concordance was observed between platforms initially, and 100% concordance was observed after retest from suspect CDV cases.”

Reviewer #2: 

1. Line 35- using a positive control plasmid 

We inserted “a” before “positive control plasmid” in line 35 to correct this.

2. Line 79- I think here it is worthwhile making a bit more of the importance of the diagnostics. As the disease clinical signs can sometimes resemble rabiles then this is an important thing to state here. Not least because diagnostics for CDV in the field will mainly be cadaver based as there is no real treatment option for CDV.

We agree and added the following to Lines 82-98 and reworked the POC section and placed that paragraph earlier to flow better in the manuscript: 

POC diagnostics can help address many challenges in understanding CDV presence, transmission patterns, identification of disease outbreaks, and conservation threats in wildlife. Some of these challenges include access to animals and opportunistic testing across small and large geographic ranges, low density or elusive behavior of some target species, and absence of or limited monitoring efforts. Others challenges researchers face include limited expertise necessary for appropriate animal sample collection, handling, and storage (including maintaining a cold chain), absence of available laboratory testing and differentiating from disease such as rabies which can present with similar clinical manifestations, and/or logistical challenges related to permit processes needed for regional or international sample shipping for laboratory testing. These challenges are compounded in remote and/or low resource settings. POC diagnostics are increasingly providing opportunities for rapid testing by researchers while they are already collecting data in the field, or when handling sick or dead wildlife, and with the development and validation of more user-friendly kits, researchers have opportunities to overcome many of these obstacles and logistical challenges that would normally impede testing.

3. Line 88- maybe have highlighted may sound better? 

We agree and have inserted “have highlighted” in line 80 before “a need for development…” to clarify. 

4. Line 99- four should be in words 

We have replaced “4” with “four” in line 115.

5. RNA extraction using the Biome RNA field prep kit- are the reagents for the different buffers- e.g. BLB available online? If so could you put a link in to these? Or are they unknown for commercial reasons?

 These buffers and their components available from Biomeme Inc. are proprietary and therefore information on the reagents is unavailable to us.

6. Line 136- a comma after expelling may help with reading flow 

We have inserted a comma after expelling in line 165.

7. You have several different extraction procedures- would this not affect levels of DNA extracted and thus results?

In this study, we calculated copy number of virus detected from duplicate samples for each of the methods of RNA extraction in order to compare their efficacy across the two platforms in question. Data was normalized to account for different elution volumes for the different RNA extraction kits. This allowed normalization and comparison of total CDV viral copy recovery within the total RNA extract across platforms. However, the extraction method used on site at the virology laboratory in Vienna, Austria, was the only protocol we could not normalize using this method because the Virology lab processed their samples differently (organ tissue suspension) from how the Biomeme samples were processed (organ swab, see methods), which is why we did not compare Ct values. 

8. In several places you say that the samples were tested in singlicate and then in the results you mention some in duplicate and some in triplicate. This was a bit unclear and confusing so maybe you could look at explaining this a little differently? It is also more common to do samples in duplicate or triplicate

We appreciate the reviewer pointing this out. All samples testing with the Biomeme two3 machine were tested in singlicate due to the limited number of 3 wells available in the machine and the need to include a positive and negative control. This is the limitation of the platform which we address in the discussion in line line 707. Where possible (when using the benchtop thermocycler), samples were run in duplicate or triplicate (Fig 3 and 4), and replicates were chosen based on what we could accommodate on our 48 well Bio-Rad plate including all controls and standards. All hair samples were run in singlicate due to the higher sample volume. 

To clarify this in the manuscript we added a sentence in line 194 of the methods section to read “All samples were tested in singlicate except in experiments where we could accommodate duplicate or triplicate replicates on the plate where indicated”.

In line 443 we also found a typo and corrected the sentence to read: “RT-qPCR was performed in singlicate and all tests were performed using Biomeme LyoRNA™ RT-PCR reagent, CDV primers and probe, on the Bio-Rad MiniOpticon. Ct values were averaged across triplicate samples and plotted with standard deviations (Fig 5A).”

9. You mention the animals which died during CDV outbreaks. Was any pathology done on these animals to confirm the disease as they could have died of anything?

Full necropsy and histopathology exams were performed on a subset of raccoon cases (n=4) from the New York outbreak to confirm disease, but additional examination was not performed on any of the other cases we screened. We have added “and in all but four cases, described below, no additional pathological examination was performed” at Line 294 and “but no additional pathological examination was performed.” at Line 314.

We also add the following to line 297: Complete necropsy examination and a full set of tissues in 10% neutral buffered formalin was collected from four raccoons that died or were euthanized during the outbreak, and histopathology examination confirmed CDV disease in these animals. 

10. Line 317- 3 should be in words 

We have replaced “3” with “three” in line 371.

11. Line 326- 27- three should be in words

We have replaced “3” with “three” in line 373.

12. and the n = 3 seems a bit repetitive? 

We think the reviewer is referring to original lines 326-327 in Figure 2. For this experiment the samples were run in singlicate, but the experiment was repeated 3 times (n=3 experiments). To provide more clarity we have therefore removed “n=3” from line 383 to reduce redundancy and this sentence now reads as: “Shown are standard deviations of the average from three independent experiments.” 

For the CDV detection comparison with Biome- were the results in Lines 356-360 significant? Again, the use of triplicates when the methodology says singular is a little confusing

Original lines 356-360 refer to Figure 4, and p-values presented in Figure 4A and in the Fig. 4 caption were statistically significant at the lower copy number concentrations, but not at 5000 copies. To add clarity, we have corrected the following sentence in lines 413-416 to read: “Samples were tested in singlicate using a CDV plasmid positive control and Biomeme LyoRNA™ RT-PCR mastermix, primers and probe. Cycle threshold (Ct) values from three independent experiments were averaged and plotted with standard deviations.”

To reduce redundancy and keep consistency with Figure 2 we have changed the following sentence in lines 431-433 to read: Standard curves of a standard dilution series of a CDV control plasmid with known copy number from triplicate experiments on the Biomeme two3™ (B) or the BioRad MiniOpticon (C).

13. Line 429- four should be in words 

We have replaced “4” with “four” in line 497.

14. Line 430- comma after graphs 

We have inserted a comma after “graphs” in line 498.

15. Line 432- 3 and 4 should be in words 

We have replaced “3” with “three” and replaced “4” with “four” in line 500.

16. Line 438- the second ‘been’ can be removed 

We have removed “been” and that part of the sentence has been rewritten to “hair has not been historically used” in line 506.

17. Line 448- 4 should be in words

We have replaced “4” with “four” in line 516.

18. Line 534- in my experience there is a lot of DNA and RNA lost using the Qiagen kits due to the use of the filters which tend to be relatively poor at releasing DNA after washing steps and Line 536- you say that this is likely due to the RNA extraction- which part of it, and could you optimise it in any way?

We added the following sentence at Line 624: “Further work is needed to determine if the reduction in viral detection is due to either reduced binding of all available RNA to the column or less efficient release of RNA off the column, and if any optimization can be done to improve recovery.”

19. Line 543- I would always avoid saying test with larger sample sizes- because it raises the simple question ‘why didn’t you do that then?’ (although I am not that cruel). Maybe state that using a variety of different viral titres would be beneficial?

We have changed the following sentence in line 632 to read: “Further testing with a variety of different viral titers will be beneficial in ascertaining how this kit performs in a variety of clinical CDV samples that contain low versus high viral loads.”

20. Line 564- and animals in a clinical setting 

We have added “and in a clinical setting” to the end of the sentence in line 673.

21. Line 573- comma after tissue types 

We have inserted a comma after “tissue types in line 681”.

22. Line 573-575- consider rewording this as two mentions of other tissue types makes it harder to follow

We have changed the following sentence in line 681 to read: “When compared to other tissues types, the Ct differences across the hair samples were comparable, suggesting hair as a good, non-invasive sample for CDV POC testing.”

23. Line 579- comma after several years

We have inserted a comma after “years” in line 687.

24. Line 588 and 589- these two sentences may benefit from being linked 4

We have linked the two sentences in line 697. The sentence now reads as: “The entire Biomeme POC platform is lightweight and very portable, and all of the equipment and supplies are able to fit in a standard sized backpack.”

25. Line 597- 5 in words 

We have replaced “5” with “five” in line 706.

26. Line 600 and 603- 3 in words 

We have replaced “3” with “three” in both instances (line 710 and 714).

27. Acknowledgements- can you please just confirm that Biomeme were not involved in the study and didn’t supply reagents etc?

We have added to acknowledgements section the following: “Biomeme Inc. was not involved in this study. All reagents and equipment using this platform were purchased from Biomeme Inc. for use in this study.”

---

## [Editor Report · Decision Letter 1]

7 Apr 2020

Development and validation of a portable, point-of-care canine distemper virus qPCR test

PONE-D-20-01230R1

Dear Dr. Seimon

We are pleased to inform you that your manuscript has been judged scientifically suitable for publication and will be formally accepted for publication once it complies with all outstanding technical requirements.

With kind regards,

Simon Russell Clegg, PhD

Academic Editor

PLOS ONE

Additional Editor Comments (optional):

Many thanks for re-submitting your manuscript to PLOS One

I have reviewed your modifications, and as you have addressed all the points raised, I have recommended your manuscript for publication

You should hear from the editorial office soon

It was a pleasure working with you and I wish you all the best for your future research

Hope you are keeping safe during this difficult time

Thanks

Simon

---

## [Editor Report · Acceptance letter]

9 Apr 2020

PONE-D-20-01230R1 

Development and validation of a portable, point-of-care canine distemper virus qPCR test 

Dear Dr. Seimon:

I am pleased to inform you that your manuscript has been deemed suitable for publication in PLOS ONE. Congratulations! Your manuscript is now with our production department. 

With kind regards,

on behalf of

Dr. Simon Russell Clegg 

Academic Editor

PLOS ONE